# Taste Receptors: The Gatekeepers of the Airway Epithelium

**DOI:** 10.3390/cells10112889

**Published:** 2021-10-26

**Authors:** Katleen Martens, Brecht Steelant, Dominique M. A. Bullens

**Affiliations:** 1Allergy and Clinical Immunology Research Group, Department of Microbiology, Immunology and Transplantation, KU Leuven, 3000 Leuven, Belgium; katleen.martens@kuleuven.be (K.M.); brecht.steelant@kuleuven.be (B.S.); 2Department of Bioscience Engineering, University of Antwerp, 2020 Antwerp, Belgium; 3Clinical Division of Pediatrics, University Hospitals Leuven, 3000 Leuven, Belgium

**Keywords:** taste receptors, respiratory tract, chronic airway diseases, polymorphisms

## Abstract

Taste receptors are well known for their role in the sensation of taste. Surprisingly, the expression and involvement of taste receptors in chemosensory processes outside the tongue have been recently identified in many organs including the airways. Currently, a clear understanding of the airway-specific function of these receptors and the endogenous activating/inhibitory ligands is lagging. The focus of this review is on recent physiological and clinical data describing the taste receptors in the airways and their activation by secreted bacterial compounds. Taste receptors in the airways are potentially involved in three different immune pathways (i.e., the production of nitric oxide and antimicrobial peptides secretion, modulation of ciliary beat frequency, and bronchial smooth muscle cell relaxation). Moreover, genetic polymorphisms in these receptors may alter the patients’ susceptibility to certain types of respiratory infections as well as to differential outcomes in patients with chronic inflammatory airway diseases such as chronic rhinosinusitis and asthma. A better understanding of the function of taste receptors in the airways may lead to the development of a novel class of therapeutic molecules that can stimulate airway mucosal immune responses and could treat patients with chronic airway diseases.

## 1. Introduction

The primary function of taste is to evaluate the nutritious content of food and to prevent the ingestion of noxious substances. Taste perception involves the recognition of five basic tastes—bitter, sweet, sour, salty, and umami—by taste cells expressing specific receptors for these tastes [1]. More specifically, tasting sweet and umami is mediated by a small family of three G-protein coupled receptors (GPCRs) (i.e., T1R1, T1R2, and T1R3). The bitter taste is mediated by a family of approximately 25 highly divergent GPCRs, namely the T2Rs. These T2Rs play an important role in the regulation of taste related behavioral responses such as spitting out, vomiting, coughing, and sneezing [2]. Finally, salty, and sour tastants are suggested to modulate taste-cell function by direct entry of Na+ and H+ through specialized membrane channels present on the apical surface of the taste cells [1]. Over the last decade, our knowledge regarding the function and location of taste receptors has grown exponentially. The expression of these receptors was initially thought to be confined to the oral cavity, but has recently been described in a range of other tissues (e.g., heart, gut, brain, nasal cavity, and lungs) and cell types (i.e., chemosensory, smooth muscle, endothelial, epithelial, and inflammatory cells) that are clearly unrelated to taste sensing and as such are involved in different functions (Table 1). In the oral gingiva, solitary chemosensory cells (SCCs) express T2Rs, which coordinate the immune response against Gram-negative bacteria [3]. In the gastrointestinal tract, enteroendocrine cells express T2Rs and T1Rs that are activated by sugars and other nutrients [4]. Upon activation, the enteroendocrine cells release hormones that can act locally as paracrine factors, neurotransmitters, and neuromodulators or act at distal sites [4]. Additionally, T1Rs (i.e., TAS1R2 + TAS1R3 heterodimer) play a role in glucose sensing and energy balancing [5], whereas umami and other amino acid receptors (e.g., TAS1R1 + TAS1R3 and mGluR1) act upon protein-related nutrients in the gastrointestinal tract to modulate, for instance, endo- and exocrine secretion [6,7,8,9]. T2Rs have also been involved in the motility of the gastrointestinal tract as well as in mounting innate immune responses against parasitic infection [10,11,12,13,14]. In the genitourinary system, T2Rs participate in spermatogenesis. Depletion of T2R in T2R5-Cre transgenic mice resulted in major spermatid loss [15] and T2R mediated bladder contraction via a reflex loop in the urethra [16]. In the cardiovascular system, T1Rs and T2Rs were found in cardiac myocytes both in animal models and in humans. Moreover, depletion of glucose resulted in an upregulation of T2R expression in cultured rat myocytes and in mouse heart [17]. Additionally, in the central nervous system, T2Rs have been identified in the brain stem, brain cells, parabrachial nucleus, and in choroid plexus, where they might play a role in the regulation of food intake and other, yet unknown, physiological functions [18,19,20].

In the airways, the T1R and T2R seem to possess many functions depending on the anatomical location of expression (i.e., in the upper or lower airways). More specifically, in the upper airways, T2R receptors are involved in neurogenic inflammation, modulation of ciliary beat frequency, and bacterial clearance [46]. These receptors function as bacterial sensors that induce a rapid and early immune response against secreted bacterial ligands [46]. In the lower airways, T2R receptors are involved in three different pathways: modulation of ciliary beat frequency, bronchial smooth muscle cell relaxation, and production of pro-inflammatory mediators by immune cells [47]. More specifically, triggering of T2Rs inhibit lung inflammation or smooth muscle contraction ex vivo and in asthma animal models [47]. Additionally, there are studies demonstrating a positive effect of T2R activators on lung function in patients with asthma (e.g., improvement in forced expiratory volume in one second [FEV1]) [48]. However, a potential beneficial role for T2R activation in humans remains low and additional clinical trials are needed to better understand the role of T2Rs in the airways.

The sweet T1R, in contrast to T2R, downregulates the secretion of antimicrobial peptides in a glucose-dependent manner [49]. This T1R mediated suppression of T2R-induced antimicrobial peptide secretion is presumably specific for the upper airways as this was not observed in human bronchial epithelial cultures stimulated with glucose or sucrose [21].

Taken together, taste receptors are involved in many different processes depending on the location and expression. Interestingly, the expression of these receptors can also be altered under several pathological conditions. Polymorphisms in these receptors are linked to several human disorders including airway diseases [25,26]. As such, the focus of this review will be on the role of taste receptors in the respiratory tract. We will describe their role in mucosal respiratory immunology in health and chronic respiratory diseases, highlight the mechanisms of action, and we discuss the therapeutic potential of taste receptors to potentially treat chronic airway disorders.

## 2. Bitter and Sweet Taste Receptors in the Respiratory Tract

Taste receptors have been recently described in extra-oral physiological systems including but not limited to the brain, gut, heart, airways, and testes [27]. The taste receptors in these organs do not mediate “taste” per se as they are not linked to neuronal perceptive pathways, but they still serve as local chemoreceptors in the body. The T1R and T2R taste receptors are GPCRs that act through second messengers including phospholipase C, cyclic AMP, and IP3-responsive mechanisms to modulate intracellular calcium levels, which can lead to the initiation of cell depolarization, bacterial killing, or even to the transmission of neural impulses to the brain [21,22].

### 2.1. Taste Receptors in the Upper Respiratory Tract

Based on different mouse and human genomic sequencing studies, it is estimated that each species has about 25 different T2R genes that constitute a multi-gene family, of which the members display an amino acid identity of 21%–90% [28]. Moreover, T2Rs feature seven transmembrane domains and have a short extracellular N-terminus. These receptors are activated by a wide array of bitter compounds including the acyl-homoserine lactones (AHL) produced by bacteria, caffeine, chloroquine, erythromycin, and denatonium [29]. The bitter taste receptors are expressed in both ciliated airway epithelial cells and in SCCs [30], where they play an important role as bacterial sensors to induce a rapid and early immune response against secreted bacterial ligands [22,23]. Activation of T2R on SCCs rapidly increases calcium release. Via gap junctions, surrounding epithelial cells are then stimulated to release antimicrobial compounds such as defensins and cathelicidins [21]. In addition, activation of SCCs in mice also results in the release of acetylcholine, which can activate trigeminal neurons [31]. However, the latter has still not yet been confirmed in humans. Activation of T2R on ciliated airway epithelial cells results in an immediate calcium-dependent increase in the production of nitric oxide (NO). NO will then penetrate bacterial membranes and damage bacterial DNA as well as increase the ciliary beat frequency (Figure 1) [21].

Intriguingly, the activation of T2R is counterbalanced by the activation of T1R in SCCs [21]. T1R is composed of a heterodimer of taste receptor member 2 (TAS1R2) and taste receptor member 3 (TAS1R3) [24] and detects sweet compounds such as glucose, and sucrose, among others. As a result of a continuous leak and re-uptake of glucose from the serum, the glucose concentration in the airways remained constant [47]. T1R are tonically activated by high levels of glucose, whereas low glucose levels inhibit their activation. It is hypothesized that any shift toward glucose depletion causes a favorable balance for T2R stimulation with subsequent activation of local defenses against the invading pathogen, resulting in decreased microbial numbers and restoration of the glucose concentrations [50]. Additionally, T1R can also be activated by D-amino acids produced by some bacterial species including *Staphylococcus aureus* [51]. In primary human sinonasal epithelial cell cultures, two types of D-amino-acids (i.e., D-Phe and D-Leu), produced by *S. aureus*, inhibited T2R-mediated signaling and defensin secretion by activating T1R2 and/or 3. Activation of T1R2 and/or 3 by *S. aureus* also enhanced epithelial cell death, even in the presence of the T2R activating compound denatonium [51]. Taken together, T1R agonists increase bacterial survival and propagation by impairing the host’s T2R mediated immune response and antimicrobial protein release (Figure 1) [51].

### 2.2. Taste Receptor Polymorphisms in the Upper Respiratory Tract

There is tremendous genetic variability in these subfamilies of taste receptors as evidenced by multiple receptor isoforms and polymorphisms that modulate receptor function [1,5]. Moreover, 89 common polymorphisms in the genes encoding for T2Rs are found in 5% or more of the general populations [52]. Some of these polymorphisms are even correlated with severity of upper airway diseases. One of the most well-studied correlation is the association between the polymorphisms in the tas2r38 gene and disease severity in chronic rhinosinusitis (CRS). CRS is a heterogenous disorder of the upper airways, characterized by mucosal inflammation in the nasal cavity and paranasal sinuses [53]. The polymorphisms in TAS2R38 present themselves as functional (PAV) and nonfunctional (AVI) T2R38 variants [54]. CRS patients that are non-functional AVI/AVI homozygotes cannot sense bitter T2R38 agonists such as phenylthiocarbamide [55]. These patients also show decreased production of NO against AHL stimulation, which is a class of signaling molecules produced by *Pseudomonas aeruginosa* and other Gram-negative bacteria [25]. Consequently, AVI/AVI homozygote CRS patients have an increased susceptibility to Gram-negative upper respiratory tract infections [25] and a higher burden of biofilm formation [56]. Finally, these patients are more likely to require surgical intervention [57,58], whereby CRS patients without nasal polyps, that are AVI/AVI homozygotes, may have worse outcomes after surgery [56]. Another example is the study of Mfuna Endam and colleagues. In their study, the authors investigated two separate Canadian CRS populations using pooling-based genome-wide association data that were screened for single nucleotide polymorphisms (SNPs). In addition to the SNPs in TAS2R38, they also found SNPs in TAS2R13 and TAS2R49 in CRS patients compared to healthy controls, conferring susceptibility to CRS [59]. The authors even showed coding SNPs in T1R genes in the Canadian CRS populations [59]. Polymorphisms in T1R have also been described in humans by other authors. For instance, individuals with a polymorphism resulting in either an isoleucine or valine have a higher risk for developing dental caries, have a higher carbohydrate intake, or have hypertriglyceridemia [60]. Unfortunately, up until now, studies focusing on the presence of polymorphisms in sweet taste receptors in patients with chronic airway diseases are very limited. As such, research must be aimed at further evaluating the correlation between genetic polymorphisms in the taste receptors and the antimicrobial response as well as screening the responses of the receptors to bacterial products. If certain polymorphisms result in the alteration of taste receptor function (i.e., overactivation or downregulation), this can be restored by applying the correct agonist or antagonist. These studies can be increasingly important considering the roles of these taste receptors in innate immunity and protection against bacterial infection in the respiratory tract.

### 2.3. Taste Receptors in the Lower Airways

Members of the T2R family have been identified in different pulmonary cell types including the airway smooth muscle, various epithelial cell types as well as hematopoietic inflammatory cells (i.e., macrophages, mast cells, lymphocytes, and neutrophils) [32,38,61,62]. Unfortunately, the presence of T1R in the lungs is not completely known (Figure 2).

The expression of T2R has been reported in the respiratory tract of different animals [45,63,64,65]. For instance, in the study of Sbarbati et al., the authors demonstrated the presence of SCCs in the larynx of rats, whereby these cells co-expressed α-gustducin and type III IP3 receptor (IP3R3), which is another key molecule in the bitter taste receptor pathway [63]. These results were later confirmed by Tizzano et al., where they demonstrated the expression of the T2R on the SCCs throughout the entire respiratory tract of rodents [64]. In the first study reporting the expression of bitter taste receptors in the human lower airways, the authors identified four different T2R subtypes, which were located on ciliated airway epithelial cells [38]. In this study, airway epithelial cells were obtained from the trachea and bronchi of human lungs and were cultured in vitro. When stimulated with denatonium, which is an agonist of T2R4, T2R10, and T2R46, increased ciliary beat frequency was observed because of increased intracellular calcium [38]. In 2010, the expression of T2R in human and mouse airway smooth muscle cells was also reported [32]. Compared to the well-known β2-adrenoreceptors, abundant transcript levels of the TAS2R10, -14 and -31 were observed and activation of these receptors with bitter compounds had a relaxant effect on the murine airway smooth muscle [32]. This effect was later confirmed in the small airways of mice [33] and in guinea pig [34]. Additionally, in human bronchi, a relaxant effect of T2R agonists was also observed [32,35,36,37]. For instance, Grassin-Delyle and colleagues characterized the relaxant response of human bronchi to several T2R agonists, suggesting a predominant role for T2R5, -10, and -14 [37] and the authors could show that stimulation of human bronchi, isolated from resected lungs of healthy individuals, with different T2R agonists, resulted in bronchial relaxation [37]. Of note, T2R receptors have also been found in smooth muscles of other organs including the gastrointestinal tract [66]. Activation of T2R receptors by denatonium resulted in region-dependent contractility changes in mouse intestinal muscle strips and in a delay in gastric emptying in vivo. These effects were the results of an increase in intracellular calcium release and in extracellular calcium influx [66]. These findings might also explain the relaxant effect of T2R agonists in the lower airways.

T1R is expressed in SCCs in the nasal cavity, but these cells decrease in number in the trachea and seem to be absent in the lower airways [64]. The study of Lasconi and colleagues also reported a role for T1Rs in the induction of a physiological response in the tracheal epithelial cells of rats [67]. More specifically, they demonstrated that stimulation of T1R on tracheal slices with the artificial sweeteners, sucralose, saccharin, and acesulfame-K, resulted in increased intracellular calcium levels [67]. Expression of T1R has also been reported in the pulmonary vasculature [45]. Stimulation of rat lung microvascular endothelial cells with one of the barrier-disruptive agents’ lipopolysaccharide (LPS), VEGF, or thrombin, decreased the expression of TAS1R3. Moreover, exposure of lung microvascular endothelial cells to sucralose attenuated LPS and thrombin-induced endothelial barrier dysfunction in vitro and prevented the formation of bacteria-induced lung edema in vivo [45]. These findings suggest that activation of T1R3 by sucralose protects the endothelium from edemagenic agent-induced barrier disruption.

Immune cells play a critical role in maintaining homeostasis in the human body by neutralizing foreign insults. Immune cells express multiple cell surface receptors including the toll-like receptors and C-type lectin receptors that recognize pathogen associated molecular patterns. Studies have now also reported the presence of T2R on tissue resident and lung infiltrating immune cells [26,39]. The presence of taste receptors on immune cells further consolidates their ability to recognize microbial products and induce a proper immune response. A genome-wide analyses of isolated blood leukocytes has demonstrated the presence of TAS2Rs at the transcript level. Moreover, the authors demonstrated an increased expression of TAS2R13, 14, and 19 in blood leukocyte samples of asthmatic patients [26]. In the same study, TAS2R agonists chloroquine and denatonium inhibited LPS-induced release of multiple pro-inflammatory cytokines and eicosanoids (i.e., TNF-α, IL-13, and MCP-1) from human blood leukocytes of adult asthmatic patients [26]. Additionally, bitter compounds including chloroquine, colchicine, and erythromycin can inhibit cytokine release from tissue macrophages, myeloid derived cell lines, circulating monocytes and macrophages/dendritic cells [40,41,42,43,44]. T2R have also been reported on T lymphocytes. Higher expression levels of TAS2R38 were reported on activated and memory T cell populations compared to naïve T cells [62]. In addition to lymphocytes, the presence of T2R have also been demonstrated on monocytic and granulocytic innate immune cells [61,68,69]. For instance, activation of TAS2Rs on isolated human peripheral blood neutrophils inhibited migration of immune cells [69]. Finally, nine distinct TAS2R isoforms have been reported on cord blood-derived mast cells, with TAS2Rs 4, 46, and 14 expressed at high levels [39].

### 2.4. Taste Receptor Polymorphisms in the Diseased Lower Respiratory Tract

Many studies have demonstrated the presence of multiple subtypes of bitter and even sweet taste receptors on airway cells in the lower airways. As mentioned previously, upregulation of TAS2R gene expression has been reported in leukocytes of subjects with severe asthma [26]. In addition, polymorphisms in the TAS2R38 gene have also been linked with patients with cystic fibrosis (CF) [70]. Moreover, the frequency of the PAV allele (i.e., the functional allele) was significantly reduced in CF patients with nasal polyps that required additional sinus surgery, and in CF patients that were colonized with *P. auruginosa* [70]. Finally, TAS2R38 polymorphisms have also been described in patients with primary ciliary dyskinesia [71] and sweet taste receptors have been implicated in acute respiratory stress syndrome [45]. These studies demonstrate that taste receptors might contribute to physiological and pathological phenotypes and that genetic variations in these receptors might contribute to disease severity and risk. The next critical step will be to determine which specific T2R or T1R receptors are most important with respect to inflammatory airway diseases. Given the strong evidence that bitter and sweet taste receptors are expressed in different airway cells and their role in modulating the function of these cells, it is logical to assume that one or more polymorphism in these genes can regulate airway functions. To further advance this field of taste receptor research in the airways, it is imperative to determine which polymorphisms are present in the taste receptors of individuals suffering from airway diseases and analyze the impact of these discrepancies on the functionality of the airway cells.

## 3. Clinical Application of Taste Receptors for the Airways

The idea of stimulating or blocking the bitter and/or sweet taste receptors to strengthen the systemic innate immune response as therapeutic strategy has gained more attention worldwide. The interest in T1Rs and T2Rs is mainly because of their quick immune response (e.g., against inhaled harmful substances and their second messenger pathways that take a few seconds to minutes to be expressed) compared to other receptors expressed on the epithelium [50,72,73]. Activation of T2Rs using bitter taste agonists have been adopted to treat inflammation-linked respiratory diseases such as asthma, chronic obstructive pulmonary disease, CRS, and allergic rhinitis [47,69,74,75]. In addition, the T1R attenuation of T2R mediated anti-microbial peptide secretion may also have important clinical relevance.

As mentioned previously, CRS is a heterogenous inflammatory disorder of the upper airways, whereby conventional management involves a high prescription of antibiotics. Consequently, this high prescription contributes to the crisis of antibiotic resistance [76]. Alternative therapeutic strategies that can eradicate infections by stimulating endogenous host defenses are therefore warranted. For instance, given the fact that T1R controls the T2R-mediated anti-microbial peptide secretion depending on the glucose concentration in the upper airways, T1R can be of utmost importance in this context. Indeed, glucose is present in the airways because it tonically leaks through the epithelium and is taken up by epithelial cells via apical glucose transporters [77,78,79]. In CRS patients, however, glucose concentration in nasal secretions is three- to four-fold higher than in healthy individuals [21], most likely because of increased leakage due to epithelial damage associated with chronic infection and/or inflammation [77]. More specifically, pro-inflammatory mediators are known to disrupt tight junctions in human sinonasal cells in vitro [80]. Consequently, the higher concentration of glucose will lead to the inhibition of T2R by activating T1R and reduced production of anti-microbial peptide, whereby bacteria can further colonize the upper airways. This hypothesis, however, needs to be further elucidated. Nevertheless, if this hypothesis is true, topical application of sweet receptor antagonists such as lactisole may restore T2R signaling and normal sinonasal responses in CRS. In addition to this hypothesis, individuals with functional TAS2R38 alleles have a reduced risk of developing CRS that require surgical intervention compared to individuals that have the nonfunctional form [56,58] and these CRS patients have improved surgical outcomes [56]. This nasal function is closely related to its function in the oral cavity and the perception of bitter and sweet taste as such. Influencing taste receptors might therefore also influence altered taste perception in CRS patients [81], which is known to be a difficult to control complaint. Indeed, in the study of Lin et al., the authors demonstrated that CRS patients experienced a bitter taste as less intense and sucrose as more intense compared to healthy controls [82]. The therapeutic potential of taste receptors has also been studied in the context of allergic rhinitis. Allergic rhinitis is a chronic airway disorder, clinically characterized by sneezing, nasal discharge, nasal obstruction, and nasal hyperreactivity [53]. In the study of Ekstedt et al., the authors investigated the involvement of T2Rs in the beneficial effect of MP-AzeFlu in the pre-contracted airway of Balb/c mice [75]. MP-AzeFlu is a drug used in the treatment of allergic rhinitis patients and consists of azelastine hydrochloride (AZE) and fluticasone propionate (FP). In their study, they demonstrated that MP-AzeFlu caused airway dilatation, comparable to the effect induced by the T2R agonist, chloroquine [75]. Additionally, in the study of Kook et al., positive effects of T2R agonists were seen in allergic rhinitis patients [83]. Moreover, when quinine, caffeine, or erythromycin was instilled into the human nasal cavity of allergic rhinitis patients, a significant decrease in nasal obstruction was seen, documented by the visual analogue (VAS) score [83]. The authors also measured the nasal cavity volume using acoustic rhinometry, before and after stimulation with the bitter agonists. These findings showed that the nasal cavity volume was significantly increased after stimulation [83]. Interestingly, when the patients were instilled with saccharin, which is a T1R agonist, there was no significant difference in nasal obstruction nor in acoustic rhinometry between pre- and post-stimulation [83].

Studies focusing on the expression and the functional role of bitter and sweet taste receptors in the lower airways have provided a unique opportunity to exploit T2Rs and T1Rs in designing new therapies for asthma. It is thought that the relaxing effect of T2Rs on airway smooth muscle can be protective in these patients. Other beneficial effects of T2Rs could be their anti-inflammatory, antiproliferative, and antifibrotic properties, which could have an impact on airway remodeling, hyperplastic growth, and inflammation associated with asthma [24]. The role of T2R in airway responsiveness has been studied in a limited number of in vivo studies. For instance, Deshpande and colleagues investigated the effect of quinine and denatonium on airway responsiveness in the ovalbumin-sensitized mouse model of bronchial hyperresponsiveness [32]. During the measurements of hyperresponsiveness, mice received the bitter tastants quinine, denatonium, or albuterol (positive control) via aerosol. Administration of these tastants resulted in the reduction in airway resistance by 50% and 57%, respectively, which was more than the effect induced by albuterol [32]. These results suggest that TAS2R agonists might be more effective than β2-adrenoreceptor agonists in asthma. In the study of Sharma et al., the effect of two TAS2R agonists on asthmatic features in a prophylactic and therapeutic mouse model was also investigated [69]. More specifically, treatment with chloroquine and quinine by aerosol attenuated features of airway remodeling including smooth muscle mass, extracellular matrix deposition, and pro-fibrotic signaling. These tastants also prevented mucus accumulation and the development of airway hyperresponsiveness in mice. Finally, chloroquine and guinine exhibited protective effects on airway inflammation by decreasing the expression of cytokines and chemokines as well as reducing the infiltration of allergen-induced immune cells into the mouse lungs. The authors suggest that the inhibition of immune cell chemotaxis is a key mechanism by which these agonists block airway inflammation and improve asthmatic features in vivo [69]. Similar results were obtained with other bitter tastants in rats [84]. For instance, caffeine prevented pulmonary cytokine production and leukocyte influx in a rat model of apnea [84], whereas erythromycin decreased LPS-induced recruitment of pulmonary neutrophils in rats [85].

Unfortunately, up until now, no clinical trials have been designed to investigate the pulmonary effects of T2Rs or T1Rs in either healthy subjects or patients with asthma. This is mainly related to the recent discovery of both receptors as a potential therapeutic target and the fact that the currently available taste receptor agonists are not sufficiently selective and may induce harmful off-target effects. However, in several “older” human studies, bitter compounds such as caffeine have been tested on lung function in asthmatic patients [86,87,88,89,90]. Caffeine, a TAS2R 7, 10, 14, 43, and 46 agonist, has the potential to induce a dose-dependent increase in FEV1 and prevent exercise induced bronchoconstriction in adult patients with asthma [86,87,88]. In young patients with asthma, caffeine intake was associated with significant improvements in forced vital capacity (FVC), FEV1, and forced expiratory flow rates [91]. The authors of a recent Cochrane Database Systematic Review therefore concluded that caffeine induced a moderate improvement in airway function in patients with asthma [89]. In preterm infants, caffeine is also used to treat apnea of prematurity (AOP), whereby it reduces AOP and mechanical ventilation and enhances extubating success [92]. However, caffeine’s mechanism of action remains to be determined. Based on the current knowledge regarding the expression of T2Rs on lung cells, this beneficial effect of caffeine could be the result of activation of T2Rs and its second messenger pathway. Similarly, oral treatment with erythromycin resulted in an increase in the histamine PC20 in asthmatic patients, an effect that might be related to its effect on T2Rs [90]. A more recent study investigated the potential of other antibiotics to activate T2Rs [93]. The authors could demonstrate in vitro that azithromycin mobilizes intracellular Ca^2+^ through activation of T2R4 [93]. Azithromycin is a macrolide, related to erythromycin, with anti-inflammatory properties that has gained a lot of interest for the treatment of asthmatic patients [94]. As such, these results suggest that azithromycin might also have an anti-microbial effect in these asthmatic patients via induction of T2R function.

### Challenges of T1R and T2R Therapies

Developing novel treatment strategies targeting the T1R and T2Rs comes with various challenges. One of the main challenges has been the lack of receptor subtype-specific agonists, especially for T2Rs. Currently available agonists activate various T2Rs, making it very challenging to examine the contribution of individual receptor subtypes in the different respiratory functions. For instance, some ligands such as flufenamic acid are known to activate one subtype of T2R expressed on airway smooth muscle cells, whereas other ligands activate more than one subtype (e.g., chloroquine, quinine, and caffeine). Since compounds of diverse classes can interact with T2Rs, this further complicates the discovery of novel ligands with a high receptor specificity. To date, the T2Rs demonstrate low affinity to the currently available bitter compounds [95]. In the search for the “ideal” compound to target T1R and/or T2R, the candidate’s agonist specificity, affinity, and selective effect on specific receptor subtypes needs to be carefully studied. Moreover, efforts are ongoing to offer structural improvements to certain well-known classes of T2R bitter agonists that could lead to a novel class of receptor subtype-specific agonists [95,96,97]. Such bitter compound derivatives with higher binding affinities for their respective T2Rs may provide less off-target effects that persist with the currently available bitter compounds. Furthermore, the vast majority of the bitter taste agonists used in preclinical studies are not ideal candidates for human trials because of the high concentration needed to fully relax the airway smooth muscle.

One of the major challenges in exploiting T1Rs is the lack of insights into different existing polymorphisms in TAS1R genes. These polymorphisms can alter T1R responses to sugars, which may play a role in susceptibility to respiratory infection [98,99,100] and could influence a patient’s response to a certain compound. Increased sugar sensitivity in the airway might lead to increased repression of SCC-mediated anti-microbial peptide secretion. A recent study in Canadian CRS patients showed allele frequency differences for 16 different SNPs in TAS1R genes compared to healthy controls [59]. Further investigation of these polymorphisms would contribute immensely to the therapeutic relevance of T1R/T2R, finally evolving toward adequate precision medicine.

## 4. Conclusions

Extra-oral T1R and T2R have sparked new interest in the last decade, following the recent discovery of their expression and physiological functions in tissues other than the oral cavity. However, their mechanism of action in the respiratory tract is still under investigation, since the signaling pathways differ from the oral cavity. Recent studies have demonstrated the expression and functional role of taste receptors on epithelial cells both in the upper and lower airways as well as on immune cells and airway smooth muscle cells in the lower airways. Polymorphisms in the receptors have even been linked to disease severity. However, further in vitro and in vivo studies are warranted to further elucidate the action pathways of these taste receptors in health and disease.

From a clinical point of view, modulation of the function of these taste receptors seems to have several potential therapeutic benefits and preliminary data in humans suggest that this class of GPCRs may be of value in the treatment of chronic inflammatory airway diseases. Therefore, we believe that growing knowledge on bitter and sweet taste receptors and their ligands, may lead to the development of novel medical interventions that can be used in the treatment of chronic airway diseases.

## Figures and Tables

**Figure 1 cells-10-02889-f001:**
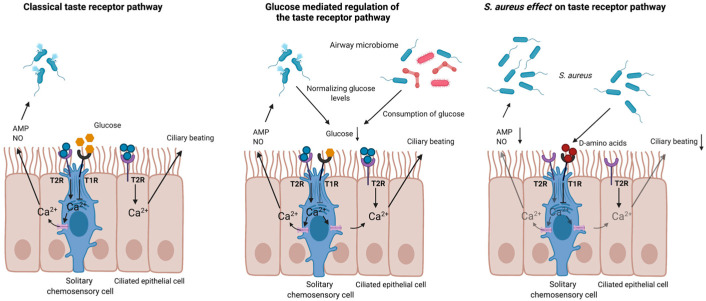
The role of taste receptors in the upper airways. T2Rs are expressed on ciliated airway epithelial cells and solitary chemosensory cells (SCCs). Activation of T2R on SCCs rapidly increases calcium release. Via gap junctions, surrounding epithelial cells are then stimulated to release antimicrobial compounds of T2R on ciliated airway epithelial cells, resulting in an immediate calcium-dependent increase in the production of nitric oxide (NO). NO will increase ciliary beating and kill bacteria. The activation of T2R is counterbalanced by the activation of T1R in SCCs. T1Rs are activated by sweet compounds such as glucose (classical taste receptor pathway). It is hypothesized that consumption of glucose by bacteria causes a favorable balance for T2R stimulation with subsequent activation of local defenses against the invading pathogen, resulting in decreased microbial numbers and restoration of the glucose concentrations (glucose mediated regulation of the taste receptor pathway). However, *S. aureus* can produce bacterial ligands such as the D-amino acids, which can activate T1Rs. The activation of T1R by these ligands blocks T2R mediated release of NO and AMP and as such, prevents killing of *S. aureus* (*S. aureus* effect on taste receptor pathway). T2R = bitter taste receptor; T1R = sweet taste receptor; NO = nitric oxide; AMP = antimicrobial peptide; Ca^2+^ = calcium. Created with BioRender.com.

**Figure 2 cells-10-02889-f002:**
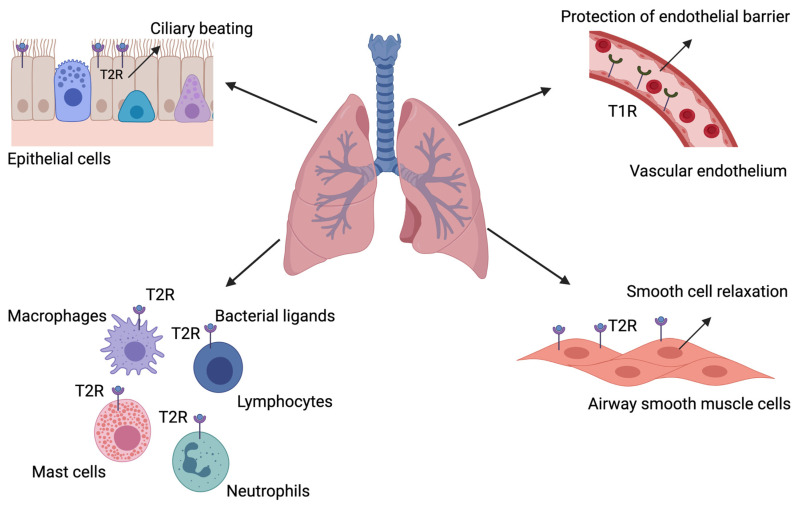
The role of taste receptors in the lower airways. T2Rs are expressed on multiple pulmonary cell types including the airway smooth muscle, epithelial cells, and immune cells (i.e., macrophages, mast cells, lymphocytes, and neutrophils). Activation of T2R on ciliated epithelial cells results in increased ciliary beating, which leads to bacterial clearance. Activation of T2Rs on various immune cells also plays an important role in bacterial clearance by inducing a proper immune response against encountered microbial products. In airway smooth muscle cells, activation of T2Rs results in bronchial smooth muscle cell relaxation. Expression of T1R has also been reported in the pulmonary vasculature. Activation of T1Rs by sweet compounds can play a role in protecting the vasculature endothelial barrier. T2R = bitter taste receptor; T1R = sweet taste receptor. Created with BioRender.com.

**Table 1 cells-10-02889-t001:** Taste receptor expression throughout the human body.

Cell Type (Organ)	Taste Receptor	Process Regulated by Taste Receptor	Ref.
**Oral cavity**			
Taste cells	T1R1, T1R2, T1R3	Tasting sweet and umami nutrients	[1]
Taste cells	T2Rs	Tasting bitter nutrients	[1]
Taste cells	Sour and salty tastants	Modulation of taste cell’s function	[1]
**Gastrointestinal tract**			
Enteroendocrine cells	T2Rs	Release of hormones as paracrine factors, neurotransmitters, and neuromodulators Motility of the gastrointestinal tract Innate immune response	[4,10,11,12,13,14]
Enteroendocrine cells	T1R and T1R3	Glucose sensing Energy balancing	[5]
Enteroendocrine cells	Umami and other amino acid receptors	Modulation of protein digestion Modulation of exocrine and endocrine secretion Metabolism and nutrient utilization	[6,7,8,9]
**Genitourinary system**			
Testis	T2R	Spermatogenesis	[13]
Bladder	T2R	Bladder contraction	[14]
**Cardiovascular system**			
Cardiac monocytes	T2R and T1R	Nutrient sensors	[15]
**Central Nervous system**			
Brain stem	T2R	Food intake	[19]
Brain cells	T2R	Food intake Other unknown physiological functions	[18]
parabrachial nucleus and horoid plexus	T2R	Assessment of cerebrospinal fluid	[20]
**Upper airways**			
Epithelial cells	T2R	Neurogenic inflammation Modulation of ciliary beat frequency Bactericidal clearance	[21,22,23]
Solitary chemosensory cells	T1R	Modulation of glucose levels Downregulation of T2R activation	[21,24]
Solitary chemosensory cells	T2R	Neurogenic inflammation Modulation of ciliary beat frequency Bactericidal clearance	[21,22,23,25,26,27,28,29,30,31]
**Lower airways**			
Airway smooth muscle cells	T2R	Bronchial smooth muscle cell relaxation Ciliary beat frequency	[32,33,34,35,36,37]
Epithelial cells	T2R	Ciliary beat frequency	[38]
Immune cells	T2R	Recognition of microbial products and induction of immune response	[26,39,40,41,42,43,44]
Microvascular endothelial cells	T1R	Protection against edemagenic agent-induced barrier disruption	[45]

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
