# Peer review of "Taste Receptors: The Gatekeepers of the Airway Epithelium"

_cells, 2021, doi:10.3390/cells10112889_

Round 1
Reviewer 1 Report
Comments:
- Please add extraoral expression of T2Rs in brain in Table 1.
- On page 7, line 258 there is a typo, it should be many not manu.
- IN lower airways role of T2Rs in Cystic fibrosis need to be added, small paragraph would be helpful.
- In the introduction part T2rs in oral innate immune response should be discussed.
Author Response
Point-to-point reply
Martens et al.: “Taste receptors: the gatekeepers of the airway epithelium.”
We would like to thank the reviewer for the positive evaluation of our manuscript. We have modified the manuscript as requested.
We have answered all questions raised by the reviewer. Please find our detailed replies to all comments below.
Comment 1: Please add extraoral expression of T2Rs in brain in Table 1
Reply: We thank the reviewer for this comment. As such, we have added the extraoral expression of T2Rs in brain in Table 1. We have also included one sentence in the introduction that mentions the expression of T2Rs in the different compartments of the central nervous system. The text now reads:
Page 2; line 59-62
“Also, in the central nervous system T2Rs have been identified, including in the brain stem, brain cells, parabrachial nucleus and in choroid plexus, where they might play a role in regulation of food intake and other, yet unknown, physiological functions (18–20).”
Comment 2: On page 7, line 258 there is a typo, it should be many not manu
Reply: We apologize for this typo. We have changed the word to many.
Comment 3: In lower airways role of T2Rs in Cystic fibrosis need to be added, small paragraph would be helpful.
Reply: We thank the reviewer for this suggestion. We have implemented a small paragraph in the review. The text now reads:
Page 8, lines 281-286
“In addition, polymorphisms in TAS2R38 gene have also been linked with patients with cystic fibrosis (CF) (70). Moreover, the frequency of the PAV allele, i.e., the functional allele, was significantly reduced in CF patients with nasal polyps that required additional sinus surgery, and in CF patients that were colonized with P. auruginosa (70). Lastly, TAS2R38 polymorphisms have also been described in patients with primary ciliary dyskinesia (71) and sweet taste receptors have been implicated in acute respiratory stress syndrome (54).”
Comment 4: In the introduction part T2Rs in oral innate immune response should be discussed.
Reply: We thank the review for this suggestion. Indeed, T2Rs are chemosensory sentinels that monitor toxic, metabolic, and infectious threats in the oral cavity and initiate a defensive response against these harmful substances. Moreover, they play an important role in the regulation of taste related behavioral responses such as spitting out, vomiting, coughing and sneezing (Harmon et al. 2021, Current Opinion in Physiology). Additionally, the group of Zheng et al. demonstrated that gingival solitary chemosensory cells express T2Rs which are involved in the immune response against Gram-negative bacteria to protect against alveolar bone loss (Zheng et al. 2019, Nature Communications).
We have implemented this information in the review. The text now reads:
Page 1, line 32-43
“These T2Rs play an important role in the regulation of taste related behavioral responses such as spitting out, vomiting, coughing, and sneezing (2). Lastly, salty, and sour tastants are suggested to modulate taste-cell function by direct entry of Na+ and H+ through specialized membrane channels on the apical surface of the taste cells (1). Over the last decade, our knowledge regarding the function and location of taste receptors has grown exponentially. The expression of these receptors was initially thought to be confined to the oral cavity but has recently been described in a range of other tissues (e.g., heart, gut, brain, nasal cavity, and lungs), and cell types (i.e., chemosensory, smooth muscle, endothelial, epithelial, and inflammatory cells), that are clearly unrelated to taste sensing and as such are involved in different functions (Table 1). In the oral gingiva, solitary chemosensory cells (SCCs) express T2Rs which coordinate immune response against Gram-negative bacteria (3).”
Reviewer 2 Report
I appreciate the opportunity to review the manuscript for publication in MDPI Cells.
The authors have highlighted taste receptors in human airways involved in the immune pathways outside the oral cavity.
I feel that the topics is an interesting and includes important area.
I have a few comments.
The authors did not mention the topics on possible relation with allergic rhinitis which is another representative upper airway atopic disease. Brief section of review would be of interest in comparison with BA.
In Table 1, “Bacterial clearance” should be bactericidal.
There are several interesting articles which are unlisted in References but useful for better understanding.
Civantos AM, Maina IW, Arnold M, et al. Denatonium benzoate bitter taste perception in chronic rhinosinusitis subgroups. Int Forum Allergy Rhinol. 2021;11:967–975.
Triantafillou V, Workman AD, Patel NN, et al. Broncho–Vaxom R (OM-85 BV) soluble components stimulate sinonasal innate immunity. Int Forum Allergy Rhinol. 2019;9:370–377.
Chen J, Larson ED, Anderson CB, Agarwal P, Frank DN, Kinnamon SC, Ramakrishnan VR. Expression of Bitter Taste Receptors and Solitary Chemosensory Cell Markers in the Human Sinonasal Cavity. Chem Senses. 2019 Sep 7;44(7):483-495. doi: 10.1093/chemse/bjz042. PMID: 31231752; PMCID: PMC7357247.
Author Response
Point-to-point reply
Martens et al.: “Taste receptors: the gatekeepers of the airway epithelium.”
We would like to thank the reviewer for the positive evaluation of our manuscript. We have modified the manuscript as requested.
We have answered all questions raised by the reviewer. Please find our detailed replies to all comments below.
Comment 1: The authors did not mention the topics on possible relation with allergic rhinitis which is another representative upper airway atopic disease. Brief section of review would be of interest in comparison with BA.
Reply: We thank the reviewer for this very interesting suggestion. As such, we have implemented information regarding the involvement of taste receptors in allergic rhinitis in the review. The text now reads
Page 9, line 340-356
“The therapeutic potential of taste receptors has also been studied in the context of allergic rhinitis. Allergic rhinitis is a chronic airway disorder, clinical characterized by sneezing, nasal discharge, nasal obstruction, and nasal hyperreactivity (39). In the study of Ekstedt et al., the authors investigated the involvement of T2Rs in the beneficial effect of MP-AzeFlu in pre-contracted airway of Balb/c mice (75). MP-AzeFlu is a drug used in the treatment of allergic rhinitis patients and consists of azelastine hydrochloride (AZE) and fluticasone propionate (FP). In their study, they demonstrated that MP-AzeFlu caused airway dilatation, comparable to the effect induced by the T2R agonist, chloroquine (75). Also, in the study of Kook et al., positive effects of T2R agonists were seen in allergic rhinitis patients (83). Moreover, when quinine, caffeine or erythromycin was instilled into the human nasal cavity of allergic rhinitis patients, a significant decrease in nasal obstruction was seen, documented by the visual analogue (VAS) score (83). The authors also measured the nasal cavity volume using acoustic rhinometry, before and after stimulation with the bitter agonists. These findings showed that the nasal cavity volume was significantly increased after stimulation (83). Interestingly, when the patients were instilled with saccharin, which is a T1R agonist, there was no significant difference in nasal obstruction nor in acoustic rhinometry between pre- and post-stimulation (83).”
Comment 2: In Table 1 “Bacterial clearance” should be bactericidal.
Reply: We apologize for this typing error. We have changed bacterial to bactericidal in table I.
Comment 3: There are several interesting articles which are unlisted in References but useful for better understanding.
Civantos AM, Maina IW, Arnold M, et al. Denatonium benzoate bitter taste perception in chronic rhinosinusitis subgroups. Int Forum Allergy Rhinol. 2021;11:967–975.
Triantafillou V, Workman AD, Patel NN, et al. Broncho–Vaxom R (OM-85 BV) soluble components stimulate sinonasal innate immunity. Int Forum Allergy Rhinol. 2019;9:370–377.
Chen J, Larson ED, Anderson CB, Agarwal P, Frank DN, Kinnamon SC, Ramakrishnan VR. Expression of Bitter Taste Receptors and Solitary Chemosensory Cell Markers in the Human Sinonasal Cavity. Chem Senses. 2019 Sep 7;44(7):483-495. doi: 10.1093/chemse/bjz042. PMID: 31231752; PMCID: PMC7357247.
Reply: We thank the reviewer for suggesting these very interesting articles. We have included these articles in the text and the references list.